# Multi-phase renal tumor segmentation with Slide-SAM assisted network

**Ruiyang Jin**[1,2]  ⓘ                                                  RYJIN@MAIL.USTC.EDU.CN

[1] *School of Biomedical Engineering, Division of Life Sciences and Medicine, University of Science and Technology of China, Hefei, Anhui, 230026, P.R.China*

[2] *Suzhou Institute for Advanced Research, University of Science and Technology of China, Suzhou, Jiangsu, 215123, P.R.China*

**Quan Quan**[3]                                                      QUANQUAN20B@GMAIL.COM

[3] *State Grid Hunan Electric Power Corporation Limited Research Institute, Changsha 410007, China*

**S.Kevin Zhou**[1,4,5]                                                  SKEVINZHOU@USTC.EDU.CN

[4] *Center for Medical Imaging, Robotics, Analytic Computing and Learning (MIRACLE), Suzhou Institute for Advance Research, USTC, Suzhou Jiangsu, 215123, China*

[5] *Key Laboratory of Intelligent Information Processing of Chinese Academy of Sciences (CAS), Institute of Computing Technology, CAS,Beijing, 100190, China*

**Editors:** Under Review for MIDL 2025

## Abstract

Non-contrast CT (Computed Tomography) scans often suffer from low tissue contrast and indistinct tumor boundaries, making accurate segmentation challenging. To address this, we propose SSA (Slide-SAM assisted network), a segmentation framework that leverages the pretrained Slide-SAM model guided by box prompts from contrast-enhanced CT. By transferring spatial priors across phases, SSA significantly improves segmentation accuracy on plain-phase images and achieves additional gains on enhanced phases. Experimental results highlight the effectiveness of combining vision foundation models with inter-phase guidance for robust medical image segmentation.

**Keywords:** Segmentation, Pretrained models.

## 1. Introduction

Renal tumors are among the most common solid malignancies in the urinary system (Ljungberg et al., 2011), and their early detection and accurate segmentation are essential for diagnosis and treatment planning (Heller et al., 2019; Ljungberg et al., 2015). Multi-phase CT imaging, especially dynamic contrast-enhanced (DCE) CT, plays a vital role in capturing tumor vascular characteristics across time (Miles et al., 2013). In contrast, non-contrast (NC) CT scans typically suffer from low tissue contrast and indistinct tumor boundaries, making precise segmentation challenging (Wang et al., 2023; Moreau et al., 2025).

Recent advances in deep learning have led to significant improvements in medical image segmentation. Models like nnUNet (Isensee et al., 2021) have achieved strong performance on high-quality DCE CT scans with clear lesion boundaries. However, their performance degrades considerably on NC CT due to the lack of contrast and structural clarity.

To address this limitation, we propose a novel segmentation framework based on the pre-trained Slide-SAM (Quan et al., 2023)(Segment Anything Model (Kirillov et al., 2023)) model, which incorporates spatial prompts derived from DCE CT to guide segmentation on NC CT. By leveraging inter-phase anatomical consistency, our method integrates structural priors from enhanced phases into a cross-phase segmentation pipeline.

We validate our approach using a private multi-phase renal tumor CT dataset. Experimental results demonstrate that the proposed method provides more stable and accurate segmentation on NC CT compared to nnUNet, demonstrating its effectiveness in low-quality imaging scenarios.

## 2. Method

The proposed network architecture comprises two main components, each performing segmentation on a different CT phase using nnUNet: one on the non-contrast phase and the other on the dynamic contrast-enhanced (e.g., arterial phase) CT image. The segmentation result of the arterial phase is denoted as $M_{p\_arterial}$.

To guide the segmentation of the non-contrast phase, we extract a 3D bounding box from $M_{p\_arterial}$, denoted as $Rec_{p\_arterial}$, using the following operation:

$$Rec_{p\_arterial} = BB(M_{p\_arterial}) \tag{1}$$

where $BB(\cdot)$ denotes the computation of a 3D bounding box from the segmentation mask. Specifically, $M_{p\_arterial}$ represents the segmentation result of the arterial phase obtained by nnUNet, $Rec_{p\_arterial}$ is the 3D bounding box extracted from the segmentation mask and used as a prompt, and $BR(\cdot)$ refers to the bounding box extraction operation.

To obtain per-slice prompts for Slide-SAM, the bounding box $Rec_{p\_arterial}$ is further divided along the axial direction according to the number of slices in the volume. This results in a set of 2D rectangular box prompts denoted as $P_{p\_arterial}$:

$$P_{p\_arterial} = Slice(Rec_{p\_arterial}) \tag{2}$$

where $Slice(\cdot)$ denotes the slicing operation that converts a 3D bounding box into slice-wise 2D rectangles.

These prompt boxes are then combined with the non-contrast CT volume $V_{plain}$ and fed into the fine-tuned Slide-SAM model to produce the final optimized segmentation result $M_{SAM-plain}$:

$$M_{SAM-plain} = SlideSAM(V_{plain}, P_{p\_arterial}) \tag{3}$$

where $SlideSAM(\cdot)$ refers to the pretrained model fine-tuned for this task.

## 3. Experiment

All experiments are conducted on a private multi-phase CT dataset including plain (non-contrast) and contrast-enhanced (arterial and venous) scans.A total of 111 cases with complete phase annotations are selected for training, while 28 additional cases are reserved for testing the fine-tuned Slide-SAM model.

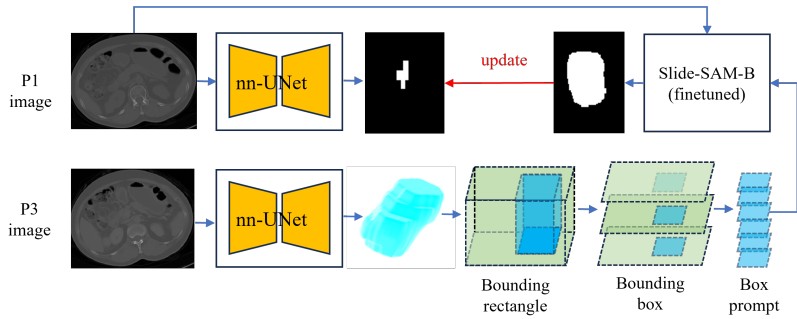

Figure 1: Framework of Slide-SAM assisted Network

Table 1 reports the segmentation performance of our proposed method (SSA) and the baseline nnUNet across three CT phases: non-contrast (NC), arterial (DCE), and venous (DCE). Under a confidence threshold of 0.8, SSA outperforms nnUNet on all phases, achieving a notable improvement on NC CT (0.771 vs. 0.742), where segmentation is most challenging.

Table 2 shows the impact of the confidence threshold, defined as the IoU between each segmentation and its box prompt. Performance peaks at 0.8, suggesting it offers the best trade-off between reliability and coverage.

Table 1: Segmentation performance on different CT phases (NC and DCE).

| Method | NC (Plain) | DCE (Arterial) | DCE (Venous) |
|---|---|---|---|
| nnUNet | 0.742 | 0.865 | 0.867 |
| SSA (Ours) | **0.771** | **0.878** | **0.885** |

Table 2: Effect of confidence thresholds on Dice score (NC phase).

| Confidence Threshold | Average Dice Score |
|---|---|
| 0.0 | 0.631 |
| 0.6 | 0.708 |
| 0.7 | 0.736 |
| 0.8 | **0.771** |
| 0.9 | 0.712 |

## 4. Conclusion

We propose SSA, a Slide-SAM assisted segmentation framework guided by box prompts from contrast-enhanced CT. SSA improves segmentation on both plain and enhanced phases, with confidence-based filtering to ensure result reliability. Experiments show clear advantages over baseline methods.

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

## Appendix A.  Data Preprocessing & implementation details

We first perform basic preprocessing on the raw kidney tumor dataset by loading each 3D CT image volume along with its corresponding segmentation mask. Based on the spatial dimensions of each volume, the axis with the shortest length (axial, coronal, or sagittal) is automatically selected as the slicing direction. A sliding window is then used to extract consecutive groups of three adjacent slices, which are stacked to form a three-channel image tensor.

For each voxel, the intensity values are clipped to a fixed Hounsfield Unit (HU) window of $[-200, 400]$ and linearly normalized to the range $[0, 255]$:

$$\mathrm{HU}_{\mathrm{clipped}} = \min(\max(\mathrm{HU}, -200), 400)$$

The normalized slices are then resized to a uniform resolution of $1024 \times 1024$ via interpolation and saved as RGB images to visually represent the selected three-slice input.

Simultaneously, the corresponding segmentation mask slices at the same spatial positions are processed identically. For each organ of interest, a binary mask is extracted. A mask channel is retained only if the organ's area in the middle slice exceeds $0.14\%$ of the entire image area. Valid masks are resized to the same standard resolution and saved as individual binary mask images.

Examples of the preprocessed images and masks are shown in Figure 2, respectively. In addition, for each image-mask pair, a metadata file is generated to store essential information including: original file path, slice index, direction index, included label indices, and corresponding organ class names.

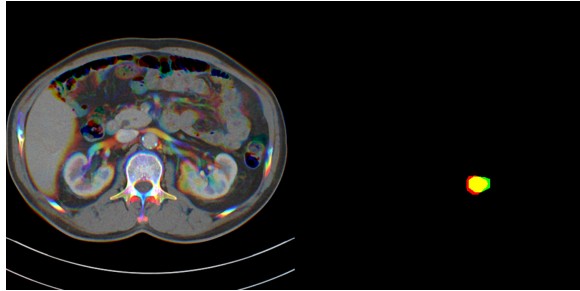

Figure 2: Example of a preprocessed 3-channel image and mask

For fine-tuning, we use the AdamW optimizer with a learning rate of 0.0002, $\beta_1 = 0.9$, $\beta_2 = 0.999$, and a weight decay of 0.1. The batch size is set to 2, and the model is trained for 100 epochs. All experiments are conducted on two NVIDIA RTX GPUs with 24 GB memory each.

## Appendix B.  Result Visualization

As shown in Figure 3, we visualize a representative case of clear cell renal cell carcinoma (ccRCC) on non-contrast CT, using the proposed Slide-SAM-based segmentation without confidence filtering (i.e., confidence $= 0$). From left to right, the ground truth, Slide-SAM prediction, and nnUNet prediction are shown in red, green, and blue, respectively.

Notably, nnUNet fails to generate valid masks for some slices (e.g., slice 1 and 8), leading to missing box prompts. However, Slide-SAM leverages its sliding-window mechanism with tri-slice input and only requires prompts for the center slice, enabling indirect inference for neighboring slices. This explains why Slide-SAM performs reasonably even without direct prompts, often outperforming nnUNet in challenging cases.

Additionally, due to the prompt propagation across slices, errors in one slice can cascade to others. For instance, segmentation inaccuracies in slice 1 can affect slices 2 and 3, highlighting the importance of inter-slice dependency, and the role of confidence in constraining segmentation quality.

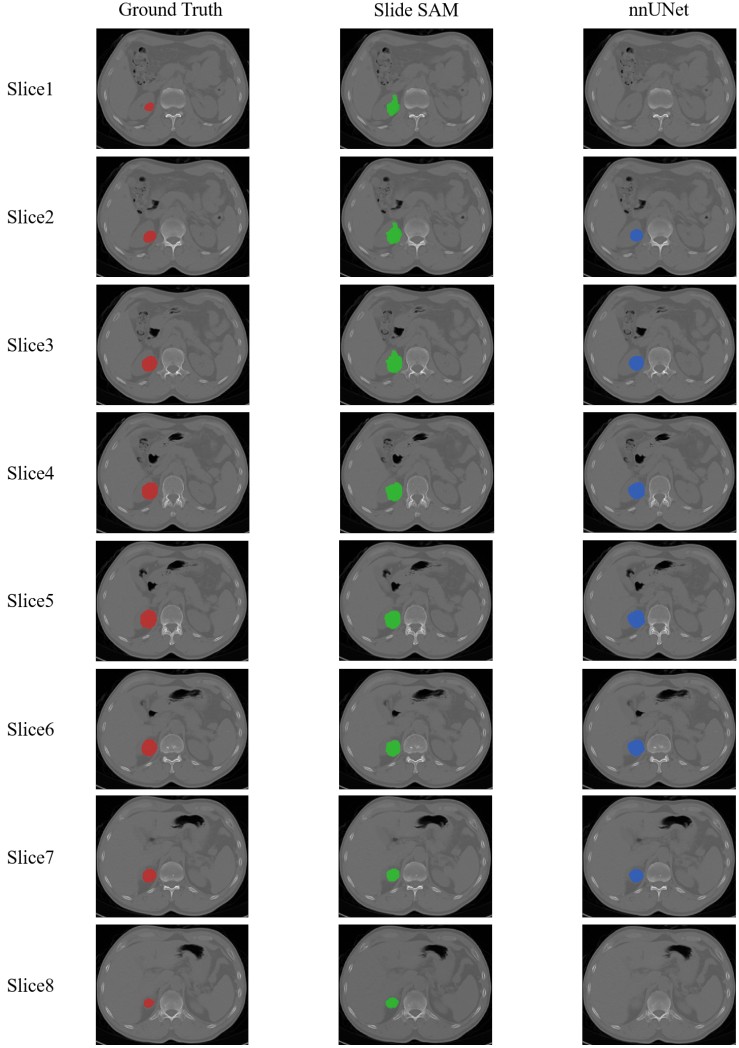

Figure 3: Segmentation results on non-contrast CT. From left to right: Ground truth (red), Slide-SAM prediction (green), and nnUNet prediction (blue).

