# OpenReview forum: "Multi-phase renal tumor segmentation with Slide-SAM assisted network"
_MIDL.io/2025/Short_Papers — MIDL 2025 - Short Papers_

### Official Review · Reviewer_avpq · 2025-04-29

**Rating:** 3
**Confidence:** 5

**Summary:**

The paper develops a nnUNet-based renal tumor segmentation framework for non-contrast CT scans, which is guided by spatial prompt from contrast-enhanced CT scans and pretrained SAM.

**Strengths:**

It uses contrast-enhanced CT to guide and enhance the segmentation performance of non-contrast CT, which can provide additional shape and boundary prior of renal tumor.

Using SAM to refine the segmentation mask is a good attempt, which explores the potential of SAM-based model in medical image applications.

The results have improvements compared to single nnUNet.

**Weaknesses:**

The motivation of the paper is not strong enough. In practical clinical situation, contrast-enhanced CT is mostly used for tumor diagnose and surgical planning, instead of non-contrast CT. So there is not a strong reason to enhance a tumor segmentation model specifically on non-contrast CT images, given that contrast-enhanced CT scans are generally available. When accuracy is required, contrast-enhanced CT can always provide a better segmentation mask compared to non-contrast CT.

Although the nnUNet is 3D model, the pre-trained SAM is still a 2D model. As 2D and 3D segmentation have a domain gap in axial spatial dependency, use a 2D mask to enhance 3D model may have some underlining issue in robustness.

---

### Decision · Program_Chairs · 2025-05-01

Accept